# Rehabilitation Program Combined with Local Vibroacoustics Improves Psychophysiological Conditions in Patients with ACL Reconstruction

**DOI:** 10.3390/medicina55100659

**Published:** 2019-09-30

**Authors:** Jung-Min Park, Sihwa Park, Yong-Seok Jee

**Affiliations:** 1Research Institute of Sports and Industry Science, Hanseo University, Seosan 31962, Korea; mine7728@hanmail.net (J.-M.P.); slim@korea.ac.kr (S.P.); 2Department of Leisure Marine Sports, Hanseo University, Seosan 31962, Korea; jeeys@hanseo.ac.kr

**Keywords:** vibration, pain, symptoms, peak torque, parasympathetic activation, range of motion

## Abstract

*Background and objective*: This study investigated the therapeutic effect of applying local body vibration (LBV) with built-in vibroacoustic sound on patients who had an anterior cruciate ligament (ACL) reconstruction. *Materials and Methods*: Twenty-four participants were randomly classified into a LBV group (LBVG; n = 11) or a non-LBV group (nLBVG; n = 13). Both groups received the same program; however, the LBVG received LBV. Psychological measures included pain, anxiety, and symptoms; physiological measures included systolic blood pressure (SBP), diastolic blood pressure, heart rate (HR), breathing rate (BR), sympathetic activation (SA), parasympathetic activation (PSA), range of motion (ROM), and isokinetic muscle strength at Weeks 0, 4, and 8. *Results*: Among the psychophysiological variables, pain, anxiety, symptoms, SBP, BR, and SA were significantly reduced in both groups, whereas HR, PSA, isokinetic peak torque (PT) of the knee joint, and ROM were significantly improved only in the LBVG. Comparing both groups, a significant difference appeared in pain, symptom, SA, PSA, isokinetic PT, and ROM at Weeks 4 and 8. *Conclusions*: The results indicate that the LBV intervention mitigated the participants’ pain and symptoms and improved their leg strength and ROM, thus highlighting its effectiveness.

## 1. Introduction

In recent years, various researchers have developed new therapeutic methods using body sounds and have employed techniques such as vibroacoustic sound therapy (VAST) [1]. VAST is a viable form of therapy since bodies are always in motion and the microvibrations that are generated in cells are energy-intensive and involved in an individual’s immune response. These microvibrations not only heal damaged cells, but also constantly express energy at the cellular level. These biological phenomena cause immediate vibrations in a human body, which can decrease recovery time from intense stress or fatigue after injury [2]. One study showed that applying vibrations for 30 min with frequency ranges of 10–100 Hz to PC12 cells, which are used to find what prion protein fragments cause neuronal dysfunction, resulted in a significantly higher neurite outgrowth due to the activation of p38 mitogen-activated protein kinases [3]. Another study confirmed the clinical efficacy of VAST when compared to laser and ultrasound therapy [4]. In that study, participants who had a heel spur were classified into two groups to observe differences in pain. One group received VAST, while the other group was treated with ultrasound and laser therapy. The results showed a tentative confirmation of the analgesic effectiveness of VAST in musculoskeletal overload conditions [4]. Another study also observed the effects of VAST among patients with various medical problems by performing a rest program for pain and symptom relief. The authors concluded that the 22-min VAST session had a 53% cumulative reduction in pain and symptoms. Other side effects such as pain, tension, fatigue, headache, and nausea were also reported to have reduced post-treatment [2].

Recently, the development of medical technology has made it easier for people to surgically treat various diseases and injuries. However, rehabilitation often takes a long time because patients feel pain and anxiety post-surgery. This is especially true for patients who have had surgery due to joint disease, and temporarily lose strength and range of motion (ROM). Although an aggressive rehabilitation program including flexibility, strength, and balance training is effective at maintaining or enhancing muscle strength and ROM [5], it may not be easy to apply such measures to patients immediately post-operation.

In general, whole body vibration (WBV) is a rather passive approach to active exercise. It has been suggested that this is similar to common types of exercise that people are accustomed to [6]. There are indications of the positive effects from WBV in immobilized patients. Moreover, VAST waves provide a similar effect to WBV [7]. Additionally, WBV induces improvements in muscular strength and performance and changes in peripheral circulation [8].

Local body vibration (LBV) devices with built-in VAST have been developed in Korea, and they are designed to be hand-operated. Unlike most sonic motors, a new device called Evocell has vibration patterns that are consistent and offer a wide range of sound waves, which change according to the flow of music. Therefore, there is a dual effect in that patients can listen to music while receiving stimuli to injured areas by the synchronized vibrations. Although this technique has been employed in a few hospitals in Korea, its effects have not yet been confirmed. LBV is designed to restore damaged tissue by transmitting vibrations and sound waves in the form of music to specific parts of the body, unlike WBV, where patients must stand on the vibrator to receive stimuli throughout the body. LBV is likely to have an additional psychological effect; however, the mechanisms are not clear.

Therefore, this study hypothesized that LBV penetrates deep into muscle tissues, affects systemic circulation and the autonomic nervous system (ANS), and potentially improves psychological condition. Consequently, such an intervention could lead to decreased pain and anxiety and greater joint flexibility and muscle strength. This study examined a group of patients with an anterior cruciate ligament (ACL) reconstruction in a randomized controlled trial and assessed the psychophysiological effect of a LBV intervention using vibroacoustic sound.

## 2. Materials and Methods

### 2.1. Study Design and Participants

This study took place in a research center from 1 December 2017 to 6 February 2018. The first assessment was conducted from 1 to 2 December 2017, the second assessment from 2 to 3 January 2018, and the last assessment from 5 to 6 February 2018. Participants were recruited through the recommendation of a surgeon familiar with LBV. Prior to the study, participants received detailed explanations regarding the study procedures and were then asked to complete questions. The included criteria required that participants underwent an ACL reconstruction no more than seven days before the start of the study. Participants were evaluated by clinical and radiological criteria, level of knee pain during the past month, and pain or difficulty in standing from a sitting position or climbing stairs [9]. All participants were patients who did not exercise regularly for over six months. Additionally, participants were also included if they had not received treatment/medication for weight loss or anything known to affect body composition, and if they did not have any internal metallic materials. The participants’ mean (SD) age was 29.25 (14.51) years. After excluding three participants (one had cardiac surgery three years prior and two refused to participate because their homes were too far from the hospital or research center) of fifty-five eligible participants, the remaining fifty-two participants belonged to one of two groups by lot and were randomly allocated to each group as shown in Figure 1. Of the 26 participants in the experimental group who were allocated to the LBV group (LBVG), one did not receive the assessment, two were lost in the follow-up phase, eight underwent arthroscopic surgery for complex injuries to their ACL or posterior cruciate ligament, and four underwent ACL arthroplasty. Therefore, eleven participants of the LBVG who underwent ACL reconstruction were analyzed in our study. Furthermore, of the 26 patients in the control group who were allocated to the non-LBV group (nLBVG), two did not receive the assessment, two were lost in the follow-up phase, and nine underwent arthroscopic surgery with a simple ACL rupture. Therefore, thirteen participants in the nLBVG who underwent ACL reconstruction were analyzed in our study. Exclusion criteria consisted of having a history of impairment of a major organ system or psychiatric disease. Participants with tumors, vascular inflammation, or kidney stones were also excluded. The LBVG received vibroacoustic pulses, whereas the nLBVG received massages by a VAST device without vibroacoustic pulses as a placebo. This study investigated psychological measures such as pain, anxiety, and symptoms; and physiological measures such as systolic blood pressure (SBP), diastolic blood pressure (DBP), heart rate (HR), breathing rate (BR), sympathetic activation (SA), parasympathetic activation (PSA), range of motion (ROM), and isokinetic peak torque (PT) in the quadriceps and hamstrings at Weeks 0, 4, and 8. Finally, twenty-four participants were interviewed and informed about the rehabilitation program they would receive. As shown in Figure 1, the participants were allocated as follows: LBVG (n = 11) and nLBV (n = 13). All participants were assigned using random number tables and assigned identification numbers upon recruitment. Participant characteristics, which indicate homogeneity, are presented in Table 1.

### 2.2. Research Ethics

This study was conducted in accordance with the Declaration of Helsinki and was approved by the Institutional Review Board at Sahmyook Univ. (2-7001793-AB-N-012018034HR). All subjects were recruited through advertisements and a written informed consent was obtained before enrollment. First, all of the subjects arrived at the research center to sign an informed consent form and to complete a self-reported questionnaire about their health status. After this procedure, all subjects participated in the experiment conducted by an expert.

### 2.3. Anthropometric Measurements

To measure body composition, all subjects were weighed while wearing light clothes and without shoes. The bioelectrical impedance analysis method was employed using BMS 330 and InBody 320 Body Composition Analyzer (Biospace Co. Ltd., Seoul, Korea), respectively. The analyzer was a segmental impedance device measuring voltage drops in the upper and lower body. Eight tactile electrodes were placed on the surfaces of the hands and feet. The precision of the repeated measurements expressed as a coefficient of variation was, on average, 0.6% for the percentage of fat mass [7].

### 2.4. Psychological Condition: Pain, Anxiety, and Symptom

A visual analogue scale (VAS) was used to measure psychological concerns. Each subject was asked to rate how they felt as they received LBV using a bipolar rating scale, which was a 5 × 10 cm bar-shaped box. The pain scale ranged from no pain (close to “0”) to severe pain (close to “10”). After participants marked within the box, a transparent paper with a score indicator was placed on top of the boxes to obtain a numerical score. The scales for measuring the levels of anxiety and symptoms were similar to the pain scale. In other words, the anxiety or symptom scale ranged from no anxiety or no symptom (close to “0”) to severe anxiety or symptom (close to “10”) [10]. Participants were evaluated by a professional psychologist at the beginning and end of the study. The reliabilities of the scales were measured by calculating Cronbach’s α, representing internal consistency. The Cronbach’s α of pain, anxiety, and symptom were 0.722, 0.816, and 0.799, respectively.

### 2.5. Physiological Conditions

#### 2.5.1. Blood Pressure and Heart Rate Measurement

The examiner placed an adjustable cuff on each participant’s upper arm and electronically pumped air into the cuff by pressing the start button on the digital BP monitor (HEM-741C-C1, Omron Co. Ltd., Seoul, Korea). SBP, DBP, and HR were measured between 1 and 2 min after the air was released automatically once a pressure of 200 mmHg was reached. Thirty minutes before the measurement, participants were not allowed to smoke or consume caffeine. During the measurement, they were placed in a sitting position with their arms raised to their heart height. If a participant’s upper arm circumference was more than 33 cm, a larger cuff was used. After two measurements, the mean value was used as data.

#### 2.5.2. Autonomic Nervous System Measurement

ANS was measured by HR variability (HRV). To obtain an accurate measure of HRV, participants were asked to refrain from exercising or consuming alcohol the day before the experiment. Food or caffeine were not consumed for 3 h prior to the measurement. On the day of measurement, HRV was tested twice at 10:00 am and 3:00 pm, and the mean values were recorded. Thirty minutes before the start of the measurement, the participants were asked to sit on a chair and rest. Then, the measurement was taken during the participants’ resting HR. HRV was measured for 2.5 min and factors affecting the measurement such as conversation, coughing, and deep breathing were controlled. uBioMacpa (Biosensecreative Co. Ltd., Seoul, Korea) and its embedded software was used for HRV measurements and analyses. HRV is the most commonly used for time domain and frequency domain analyses [11,12]. In the time domain analysis, the interval between normal QRS complexes or the instantaneous HR at a specific point is measured to calculate a normal-to-normal interval (NN interval) between consecutive normal QRS complexes. The standard deviation of NN interval (SDNN) can be determined using the NN interval. SDNN reflects all the cycle elements contributing to HRV. A decrease in SDNN means that the ability to cope with stress is reduced, and the overall health status and ANS control capacity are impaired. The RMSSD (square root of mean squared differences of successive NN intervals) is the square root of the square mean of successive NN interval differences, reflecting the short-term change in HR. This correlates well with the high-frequency index values of the frequency domain analysis and mainly reflects parasympathetic activity. When parasympathetic activity decreases, the RMSSD value also decreases. In general, the larger the RMSSD value, the more physiologically healthy and relaxed one is. Frequency domain analysis of HRV provides information on the HRV spectral components, high frequency, and low frequency range, which reflect the sympathetic and parasympathetic nervous system branches [11,12]. This spectral component is calculated mainly through short-term measurements (i.e., 2–5 min).

#### 2.5.3. Range of Motion Measurement

The ROM of an operated knee joint with an ACL reconstruction was measured using a goniometer in the extended and flexed positions [13]. During the measurement, the ROM of the injured side was measured twice when the participants were actively extended or flexed. Then, the mean values of the two measurements were recorded.

#### 2.5.4. Isokinetic Strength Measurement

Participants were positioned in the isokinetic dynamometer (HUMAC^®^/NORM™ Testing and Rehabilitation System, CSMi, Stoughton, MA, USA) according to the manufacturer’s guidelines for evaluating knee flexion/extension. Each participant sat in an adjustable seat and the tested limb was fixed in place with a Velcro strap on a support over the quadriceps while the knee joint was positioned at their flexed knee angle. The testing apparatus was set up and the participants were positioned and stabilized uniformly while sitting. Testing was performed on the uninjured side first and then performed on the operated side to diminish the possible apprehension of the participants. The designated movement patterns were assessed at a 60°/s concentric protocol. The data were evaluated and analyzed on the operated side. Before each joint measurement, participants were allowed to practice the movement pattern four times as they preferred to become familiar with the movement before performing five maximal test repetitions. All tests were supervised by one trained expert [14].

### 2.6. Rehabilitation Program and Local Body Vibration Administration

Participants completed a supervised progressive rehabilitation program for eight weeks (Table 2). All participants agreed not to change their daily activity patterns—outside of their participation in this study—or their dietary habits.

Participants performed workout sessions with Evocell (Shinoo Medison, Gangwon-do, Korea; Figure 2). The vibration patterns of Evocell are consistent and have a wide range of sound waves that change frequency according to the flow of music. This device has a maximal impulse of 1658 Hz, which has been developed to reduce pain by delivering stimuli below the skin. To reduce the friction of skin caused by equipment and to relieve pain, an exogenous gel (menthol-based material) was used with a starting impulse intensity at 20% of the maximum impulse for 30 min, similar to protocols from Lundeberg and colleagues [15].

The goal of the first stage focused on decreasing pain, tolerating weight bearing, and improving ROM and gait pattern. The goal of the second stage focused on tolerating full weight bearing, increasing passive ROM, and improving neuromuscular control. Finally, the goal of the third stage focused on increasing leg strength, maintaining balance ability, and improving proprioception. The rehabilitation exercises, along with their repetitions and the sets used in this study, were extracted from a study and applied as a method of managing knee problems and preventing reinjury [14].

### 2.7. Data Analysis

All data were reported as mean (SD). The sample size was determined using G*Power v 3.1.3, considering an a priori effect size f^2^(*V*) = 0.35 (medium size effect), α error probability = 0.05, and power (1—β error probability) = 0.95. Although a sample size of 30 was recommended, the current sample included 24 participants. Based on the results of the Kolmogorov–Smirnov test, the non-parametric Mann–Whitney U test and Friedman test were used to examine the differences of variables between groups and to examine the changes of variables among times. Significance was set at *p* < 0.05. SPSS version 18.0 (SPSS Inc., Chicago, IL, USA) was used for all analyses.

## 3. Results

### 3.1. Effect of Local Body Vibration on Psychological Condition

The rehabilitation program applied in this study improved the psychological measures of LBVG and nLBVG. As shown in Table 3, although the anxiety was not significantly different between the groups at all times, the pain of the LBVG was significantly lower than that of the nLBVG at Week 8 and the symptoms of the LBVG were significantly lower than those of the nLBVG at Weeks 4 and 8. In other words, a significant effect of the LBV-intervention concerning pain and symptoms was evident.

### 3.2. Effect of Local Body Vibration on Physiological Condition

#### 3.2.1. Effect of Local Body Vibration on Cardiorespiratory Variables

The rehabilitation program applied in this study somewhat improved the cardiorespiratory measures of the LBVG and nLBVG. As shown in Table 4, SBP, HR, and BR were significantly changed in the LBVG, whereas only SBP and BR were significantly changed in the nLBVG from the baseline to the end of the experiment. However, all of the cardiorespiratory measures were not significantly different between the groups at all times. In other words, a significant effect of the LBV-intervention was not found concerning the cardiorespiratory variables of physiological condition.

#### 3.2.2. Effect of Local Body Vibration on Autonomic Nervous System

The rehabilitation program in this study somewhat improved the measures of autonomic nervous system in the LBVG and nLBVG. As shown in Table 5, SA and PSA were significantly changed in the LBVG, whereas only SA was significantly changed in the nLBVG from the baseline to the end of the experiment. Specifically, the SA measure of LBVG was significantly different when compared with that of the nLBVG at Week 4. The PSA measure of LBVG was significantly higher than that of the nLBVG at Weeks 4 and 8. In other words, a significant effect of the LBV-intervention was found concerning a decreased SA and increased PSA of physiological condition.

#### 3.2.3. Effect of Local Body Vibration on Strength and Range of Motion

The rehabilitation program in this study improved the measures of isokinetic strength and ROM only in the LBVG. As shown in Table 6, the PTs of the extensor and flexor were significantly changed in the LBVG, but there were no significant changes in the nLBVG from the baseline to the end of the experiment. Specifically, the extensor PT of the LBVG was significantly higher than that of the nLBVG at Week 8. The flexor PTs of the LBVG were also significantly higher than those of the nLBVG at Weeks 4 and 8. The ROM of the LBVG was also significantly higher than those of the nLBVG at Weeks 4 and 8. In other words, a significant effect of the LBV-intervention was found concerning increased isokinetic strength and improved ROM of the operated knee joint.

## 4. Discussion

This study provided evidence that the LBV application on an operated knee joint can improve the patients’ psychological condition and physiological capacity including reduced pain and symptoms, improved ANS activity, and increased leg strength and ROM. These positive results may be related to a neurological effect through listening to music and to an improvement in ANS through sound stimuli. Specifically, an increase in vascular circulation during the rehabilitation program played a key role in the healing of the operated knee joint.

Petrofsky and Lee reported that blood circulation plays a vital role in tissue healing and that a richly vascularized area heals faster than a poorly vascularized area [16]. Muscle contractions evoked by imposed vibrations on the body can induce changes in peripheral circulation [13]. In fact, music produces a certain amount of vibration. Objective research has validated the healing effects of combined music and vibrations [17]. In VAST, the relaxing effect of music on the mind is amplified by the relaxing effect that acoustic vibromassage has on the body, producing a deep state of relaxation. VAST converts musical melodies and rhythms into waves that can be felt within a patient’s body [17,18]. VAST has long been recognized as a more complex intervention than music alone, and it uses physical stimuli in the form of a pulsed sinusoidal low frequency wave to produce mechanical vibrations that are applied directly to the body [19].

Lane and colleagues reported that the neural correlates of vagal tones developed from the tenth cerebral neuron involve mental stresses that included cognitive and emotional elements such as surgery. The vagal (high frequency) component of HRV predicts survival in patients with post-myocardial infarction, and it reflects the vagal antagonism of sympathetic influences. The results of their research showed the neural correlates of vagal-HRV during various emotional states in real time. They correlated vagal-HRV with measures of regional cerebral blood flow derived from positron emission tomography and 15 O-water in 12 healthy women during diverse emotional states [11]. The heart is dually innervated by the ANS such that relative increases in SA are associated with HR increases and relative increases in PSA are associated with HR decreases; thus, relative sympathetic increases cause the time between heart beats (inter-beat interval) to become shorter and relative parasympathetic increases cause the interval between beats to become longer [11]. One way to index the central control of the heart via the vagal nerve is the use of HRV [20]. Consequently, HRV largely reflects the respiratory gating of the output of the vagal nerve on the sinoatrial node of the heart [21,22,23].

In this study, psychological problems (i.e., pain, anxiety, and symptoms) from surgery and recovery time improved in the nLBVG and LBVG from the baseline to Week 8. However, the pain and symptoms improved more in the LBVG when compared with the nLBVG at Week 8, highlighting the effects of the rehabilitation program. These results are comparable to those of Lane and colleagues [11]. Moreover, we think that the improvement of psychological problems was affected by ANS, but not by the cardiorespiratory system. As shown in the results of this study, although SBP, HR, and BR were significantly lowered in the LBVG, these cardiorespiratory measures were not significantly different between LBVG and nLBVG from the baseline to the end of the experiment. However, significant effects of the LBV-intervention were found concerning decreased SA and increased PSA of physiological condition in the LBVG. In other words, the positive changes in ANS contributed to improved psychological stability, which may have led to the reduction of pain and symptoms.

Similarly, Lundeberg and colleagues reported that vibrations reduced acute or chronic pain that had a musculoskeletal origin [15]. Moreover, they argued that various modes of electrical stimulation could be used as alternatives to analgesics and surgical procedures. In fact, many authors have suggested that stimulation activates the natural self-healing abilities of the body and its capacity to recover as well as bringing the body and mind into balance, which can reduce inflammation and chronic pain [2,24,25,26]. Warth and colleagues investigated nine participants with advanced cancer who took part in single-sessions of music therapy, lasting for 30 min [27]. From their results, VAS was feasible; although, a graphical and statistical examination revealed only marginal mean changes between pre- and post-testing. While HRV parameters differed between individuals, mean changes over time remained relatively constant. Examination of individual trajectories revealed that vibroacoustic stimulation may impact the autonomic response. The results above are very similar with the results of our study.

Many patients with osteoarthritis are seeking help with disease management from alternative therapy. When used along with allopathic medicine, these types of therapy may increase a patient’s quality of life [28]. The music vibrations harmoniously combine to produce a new type of relaxation for patients. Lundqvist and colleagues reported that vibroacoustic sound waves reduce anxiety and aggression in some patients, thus reducing the need for medical treatment [29]. Koike and colleagues reported that VAST improved the psychological symptoms of 15 nursing home residents with symptoms of depression [30]. Those participants received VAST for 30 min per day for 10 days, and their depression levels were measured using the dementia mood assessment scale (DMAS). Tympanic temperature, pulse, BP, and SpO2 were also measured as physiological indexes of relaxation. Based on DMAS scores, depression was mitigated after participants received VAST. Moreover, significant decreases in tympanic temperature and pulse were observed post-treatment. In their conclusion, VAST provided relaxation effects and reduced depressive symptoms for older patients.

The current study supports and validates the efficacy of VAST, providing additional evidence for the value of these therapies for acute patients who have had ACL reconstruction surgery. The application of LBV improved the patients’ muscle strength and ROM as well as fostering psychological stability. Concerning leg strength, extensor and flexor PT were significantly increased only in the LBVG; moreover, the extensor PT of the LBVG was higher than that of the nLBVG at Week 8 and flexor PTs of the LBVG were higher than those of the nLBVG at Weeks 4 and 8. These results were similar for ROM. In other words, the results showed that the vibroacoustic intervention strengthened the quadriceps and hamstrings and expanded the ROM of the operated knee joint. The strength of the tonic vibration reflex tends to increase with increasing muscle length, thus yielding an elastic resistance to slow stretch, similar to the type of resistance encountered in weaker muscles [26,31]. This explanation that the vibration-induced afferent discharge increases with muscle length has been used by many scholars [3,5,6].

In one study, massages using music vibrations were effective in facilitating the release of by-products produced by cells, thereby enabling muscles and organs to improve their function [32]. The development of the muscular vibration provides a means for studying the role played by muscle afferents in motor control, and the technique seems to have therapeutic applications. In this study, not only was LBV highly motivational and relaxing for patients, it also encouraged self-confidence, reduced pain and symptoms, and increased leg strength and ROM. Ultimately, LBV was an effective treatment for pain and symptom management, muscle activation, and ROM expansion in patients who recently underwent an ACL reconstruction operation.

## 5. Conclusions

This study confirmed that the LBV built-in vibroacoustic sound on an ACL reconstruction could improve patients’ psychological condition including pain and symptoms and physiological capacity including ANS activity. These positive changes in psychophysiological conditions may lead to increased leg strength and ROM. Eventually, the results may be related to a neurological effect through listening to music and to an increase in vascular circulation during the rehabilitation program, playing a key role in the healing of the ACL reconstruction of the knee joint.

## Figures and Tables

**Figure 1 medicina-55-00659-f001:**
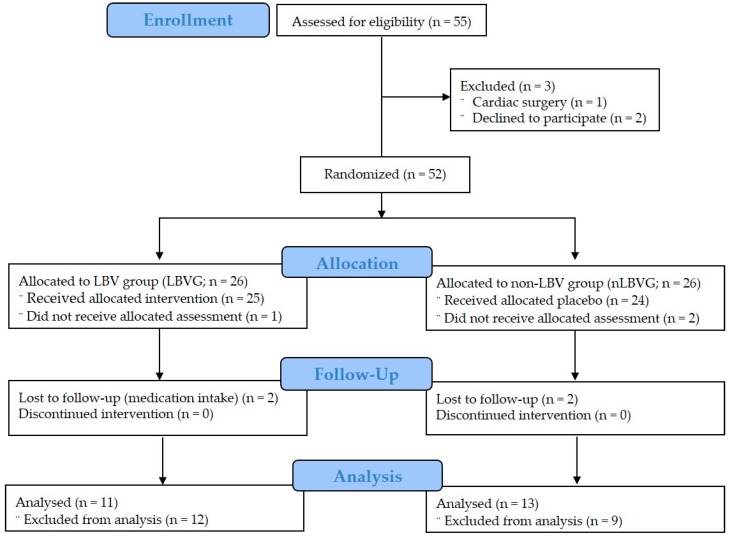
Participant allocation (consolidated standards for reporting of trials flow diagram).

**Figure 2 medicina-55-00659-f002:**
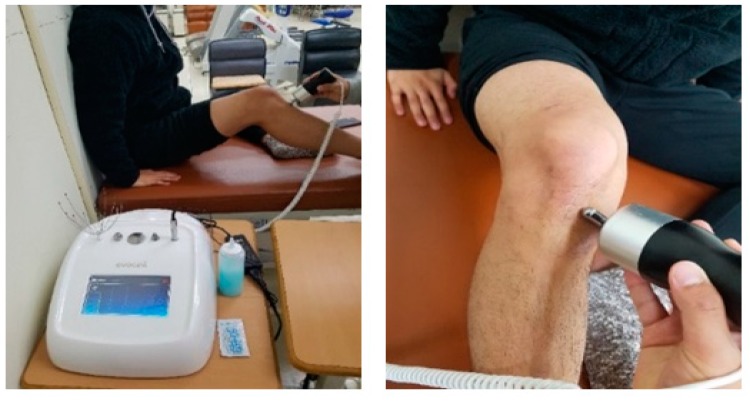
Vibroacoustic intervention applied on an anterior cruciate ligament reconstruction.

**Table 1 medicina-55-00659-t001:** Characteristics of the participants.

Items	Groups		Z	*p* *
	LBVG (n = 11)	nLBVG (n = 13)		
Age, year	26.82 ± 12.08	31.31 ± 16.49	−0.150	0.910
Height, cm	173.00 ± 4.02	174.23 ± 2.35	−0.703	0.494
Weight, kg	74.00 ± 5.37	75.50 ± 6.54	−0.321	0.776
Muscle mass, kg	36.04 ± 5.99	36.94 ± 4.26	−0.933	0.361
Fat mass, kg	14.09 ± 5.38	15.38 ± 3.60	−1.341	0.186
Body mass index, kg/m^2^	24.72 ± 1.57	24.85 ± 1.72	−0.349	0.733

All data represent the mean ± standard deviation. LBVG and nLBVG mean local body vibration group and non-local body vibration group, respectively. Symbol * was analyzed by the Mann–Whitney U test.

**Table 2 medicina-55-00659-t002:** Rehabilitation program for the local body vibration group.

Type (Length)	Program Type	Intensity/Time
Warm-up (1 day–8 weeks)	Cycling	RPE 13/15 min
	Upper and lower leg stretching	RPE 13/20 min
1st workout phase (Week 0–2)	Vibroacoustic intervention (331.6 Hz)	20% × 1658 Hz × 30 min
	Q-set	5 s × 10 reps × 3 sets
	Straight leg raise	12 reps × 3 sets
	Ball adduction	6 s × 10 reps × 3 sets
	Half squat	10 reps × 3 sets
	Weight shift	15 reps × 3 sets
	Weight bearing	15 reps × 3 sets
	Calf raise	10 reps × 3 sets
	Balance on floor	2 min × 3 sets
2nd workout phase (Week 3–5)	Vibroacoustic intervention (663.2 Hz)	40% × 1658 Hz × 30 min
	Q-set	10 s × 10 reps × 3 sets
	Straight leg raise	15 reps × 3 sets
	Ball adduction	8 s × 10 reps × 4 sets
	Leg press	10 reps × 3 sets
	Leg extension	15 reps × 3 sets
	Leg curl	15 reps × 3 sets
	Half squat	12 reps × 3 sets
	Calf raise	12 reps × 3 sets
	Balance on pad	4 min × 3 sets
3rd workout phase (Week 6–8)	Vibroacoustic intervention (994.8 Hz)	60% × 1658 Hz × 30 min
	Ball adduction	10 s × 10 reps × 4 sets
	Leg press	12 reps × 3 sets
	Leg extension	15 reps × 3 sets
	Leg curl	15 reps × 3 sets
	Half squat	15 reps × 3 sets
	Calf raise	15 reps × 3 sets
	Balance on air-pad	6 min × 3 sets
Cool-down (1 day–8 weeks)	Upper and lower leg stretching	RPE 13–15/20 min
	Icing	10 min

Note: RPE, ratings of perceived exertion or Borg scale.

**Table 3 medicina-55-00659-t003:** Differences and changes in psychological scales.

Item (Points)		Groups		
	Week	LBVG (n = 11)	nLBVG (n = 13)	Z (*p*) *
Pain	0	7.56 ± 1.21	7.70 ± 1.58	−0.511 (0.649)
	4	4.10 ± 1.44	4.53 ± 1.20	−0.503 (0.649)
	8	2.13 ± 0.90	4.54 ± 1.33	−3.661 (0.001)
	*X*^2^ (*p*) **	20.150 (0.001)	17.280 (0.001)	
Anxiety	0	6.39 ± 1.09	6.10 ± 1.24	−1.000 (0.361)
	4	3.72 ± 0.64	4.08 ± 0.95	−1.193 (0.277)
	8	3.01 ± 0.91	3.58 ± 0.95	−1.230 (0.252)
	*X*^2^ (*p*) **	17.714 (0.001)	17.522 (0.001)	
Symptom	0	7.04 ± 1.46	7.12 ± 1.89	−0.353 (0.733)
	4	3.50 ± 1.16	6.71 ± 1.10	−4.119 (0.001)
	8	2.31 ± 0.90	3.95 ± 1.21	−3.090 (0.002)
	*X*^2^ (*p*) **	18.558 (0.001)	12.875 (0.002)	

All data represent the mean ± standard deviation. LBVG and nLBVG are the local body vibration group and non-local body vibration group, respectively. Symbols * and ** were analyzed by the Mann–Whitney U test and the Friedman test, respectively.

**Table 4 medicina-55-00659-t004:** Differences and changes in cardiorespiratory variables.

Item (Units)		Groups		
	Week	LBVG (n = 11)	nLBVG (n = 13)	Z (*p*) *
Systolic blood pressure (mmHg)	0	135.09 ± 5.17	134.69 ± 12.59	−0.351 (0.733)
	4	127.09 ± 12.99	126.23 ± 7.17	−0.350 (0.733)
	8	123.45 ± 5.63	124.31 ± 4.85	−0.235 (0.820)
	*X*^2^ (*p*) **	13.905 (0.001)	7.714 (0.021)	
Diastolic blood pressure (mmHg)	0	80.55 ± 7.15	81.38 ± 12.49	−0.991 (0.331)
	4	79.73 ± 7.73	79.08 ± 11.11	−0.872 (0.392)
	8	79.09 ± 7.11	81.15 ± 11.01	−1.546 (0.134)
	*X*^2^ (*p*) **	0.800 (0.670)	0.792 (0.673)	
Heart rate (beats/min)	0	77.09 ± 6.47	71.08 ± 8.30	−1.887 (0.063)
	4	73.45 ± 6.22	78.62 ± 9.99	−1.457 (0.150)
	8	69.09 ± 7.71	73.69 ± 7.22	−1.484 (0.150)
	*X*^2^ (*p*) **	11.538 (0.003)	3.720 (0.156)	
Breathing rate (reps.)	0	29.61 ± 7.16	27.65 ± 5.54	−1.251 (0.228)
	4	19.85 ± 4.19	19.14 ± 3.06	−0.611 (0.569)
	8	19.60 ± 4.34	18.94 ± 3.48	−0.669 (0.531)
	*X*^2^ (*p*) **	13.818 (0.001)	16.769 (0.001)	

All data represent the mean ± standard deviation. LBVG and nLBVG are the local body vibration group and non-local body vibration group, respectively. Symbols * and ** were analyzed by Mann-Whitney U test and Friedman test, respectively.

**Table 5 medicina-55-00659-t005:** Differences and changes in autonomic nerve system variables.

Item		Groups		
	Week	LBVG (n = 11)	nLBVG (n = 13)	Z (*p*) *
Sympathetic activation	0	7.61 ± 0.72	7.23 ± 0.51	−1.352 (0.186)
	4	6.32 ± 0.63	7.69 ± 1.15	−3.023 (0.002)
	8	5.77 ± 0.57	6.46 ± 1.33	−1.257 (0.228)
	*X*^2^ (*p*) **	14.727 (0.001)	7.882 (0.019)	
Parasympathetic activation	0	3.55 ± 0.40	3.42 ± 0.38	−0.849 (0.424)
	4	4.49 ± 0.36	3.58 ± 0.40	−4.003 (0.001)
	8	4.96 ± 0.28	3.50 ± 0.46	−4.162 (0.001)
	*X*^2^ (*p*) **	22.000 (0.001)	2.923 (0.232)	

All data represent the mean ± standard deviation. LBVG and nLBVG are the local body vibration group and non-local body vibration group, respectively. Symbols * and ** were analyzed by the Mann–Whitney U test and Friedman test, respectively.

**Table 6 medicina-55-00659-t006:** Differences and changes in strength and range of motion in the knee joint.

Item (Units)		Groups		
	Week	LBVG (n = 11)	nLBVG (n = 13)	Z (*p*) *
Extensor peak torque (Nm)	0	77.82 ± 12.91	70.00 ± 6.53	−2.311 (0.022)
	4	123.09 ± 35.18	100.15 ± 33.83	−1.574 (0.119)
	8	178.73 ± 27.45	110.23 ± 53.88	−3.229 (0.001)
	*X*^2^ (*p*) **	13.273 (0.001)	1.385 (0.500)	
Flexor peak torque (Nm)	0	52.36 ± 5.24	47.08 ± 10.47	−1.196 (0.252)
	4	63.91 ± 6.24	43.08 ± 16.27	−3.286 (0.001)
	8	68.00 ± 10.26	50.23 ± 19.43	−2.052 (0.041)
	*X*^2^ (*p*) **	17.636 (0.001)	3.231 (0.199)	
Range of motion (°)	0	94.55 ± 16.05	86.00 ± 15.35	−1.279 (0.207)
	4	121.73 ± 14.36	89.62 ± 17.47	−3.604 (0.001)
	8	127.82 ± 12.46	92.38 ± 27.65	−2.970 (0.002)
	*X*^2^ (*p*) **	19.860 (0.001)	1.385 (0.500)	

All data represent the mean ± standard deviation. LBVG and nLBVG are the local body vibration group and non-local body vibration group, respectively. Symbols * and ** were analyzed by Mann-Whitney U test and Friedman test, respectively.

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
