# Peer review of "Rehabilitation Program Combined with Local Vibroacoustics Improves Psychophysiological Conditions in Patients with ACL Reconstruction"

_medicina, 2019, doi:10.3390/medicina55100659_

Round 1

Reviewer 1 Report

This RCT has limitations that the authors need to address.

1 Dividing the knee surgery groups between ACL reconstruction and knee arthroscopy opens the study up to selection bias.  Arthroscopy can involve minor procedures or relaively major procedures (eg chondroplsty) which cause prolonged rehabilitation compared to simple menisectomy. It would have been much "cleaner" to have only 1 surgical procedure included. the reader needs to know what arthroscopic procedures were done.

2 ACL recon is a major operation  compared to arthroscopy and so the power may be diminished as a result.

3 In my opinion there should have been 2 other arms to the study to convince me that VAST makes a difference - an arm with vibration and no music and an arm with "conservative " rehab ie no music or vibration.

Author Response

Answers to reviewer’s comments 

Thank you for your kind advice and comments for publication in Medicina. We revised the manuscript as per your comments. We represented the specific modifications in response to the comments by blue-letters in our manuscript. We sincerely appreciate your comments because your comments make our manuscript better.

Reviewer 1:

#1. Comments and Suggestions: Dividing the knee surgery groups between ACL reconstruction and knee arthroscopy opens the study up to selection bias. Arthroscopy can involve minor procedures or relaively major procedures (eg chondroplsty) which cause prolonged rehabilitation compared to simple menisectomy. It would have been much "cleaner" to have only 1 surgical procedure included. the reader needs to know what arthroscopic procedures were done. ACL recon is a major operation compared to arthroscopy and so the power may be diminished as a result.

#1. Response: Thank you for what the reviewer has pointed out above comments. In view of your comments, we selected only patients with ACL reconstruction as our subjects. Therefore, 13 patients of control group and 11 patients of experimental group were analyzed and then the results and discussion were reinterpreted according to the analysis results. In addition, we changed the title from “Influences of Rehabilitation Program Combined with Local Vibroacoustics on Patients’ Psychophysiological Conditions after Knee-joint Surgery” to “Rehabilitation Program Combined with Local Vibroacoustics Improves Psychophysiological Conditions in Patients with ACL Reconstruction”.

For example,

“Abstract: Background and objective: This study investigated the therapeutic effect of applying local body vibration (LBV) with built-in vibroacoustic sound on patients who had an anterior cruciate ligament (ACL) reconstruction. Materials and Methods: Twenty-four participants were randomly classified into a LBV group (LBVG; n = 11) or a non-LBV group (nLBVG; n = 13). Both groups received the same program; however, the LBVG received LBV. Psychological measures included pain, anxiety, and symptoms; physiological measures included systolic blood pressure (SBP), diastolic blood pressure, heart rate (HR), breathing rate (BR), sympathetic activation (SA), parasympathetic activation (PSA), range of motion (ROM), and isokinetic muscle strength at Weeks 0, 4, and 8. Results: Observing with changes of the psychophysiological variables, the pain, anxiety, symptoms, SBP, BR, and SA were significantly reduced in both groups, whereas the HR, PSA, isokinetic peak torque (PT) of knee joint, and ROM were significantly improved only in the LBVG. Comparing both groups, a significant different effect was appeared in pain, symptom, SA, PSA, isokinetic PT, and ROM at Week 4 or 8. Conclusions: The results indicate that the LBV intervention mitigated participants’ pain and symptoms and improved their leg strength and ROM, thus highlighting its effectiveness.”

For another example,

"2.1. Study Design and Participants

...........  Participants’ mean (SD) age was 29.25 (14.51) years. After excluding three participants (one had cardiac surgery three years prior and two refused to participate because their homes were far from the hospital or research center) of fifty-five eligible participants, the remaining fifty-two participants belonged to one of two groups by lot and were randomly allocated to each group as shown in Figure 1. Of the 26 patients in the experimental group which allocated in the LBV group (LBVG), one did not receive the assessment, two lost in the follow-up phase, eight underwent arthroscopic surgery for complex injuries to their ACL or posterior cruciate ligament, and four underwent ACL arthroplasty. Therefore, eleven participants of the LBVG which underwent ACL reconstruction were analyzed in our study. Further, of the 26 patients in the control group which allocated in the non-LBV group (nLBVG), two did not receive the assessment, two lost in the follow-up phase, and nine underwent arthroscopic surgery with simple ACL rupture. Therefore, thirteen participants of the nLBVG which underwent ACL reconstruction were analyzed in our study. Excluded criteria had any history of impairment of a major organ system and psychiatric diseases. Participants with tumors, vascular inflammation, or kidney stones were also excluded. The LBVG received vibroacoustic pulses, whereas the nLBVG received massages by a VAST device without vibroacoustic pulses as a placebo. This study investigated psychological measures such as pain, anxiety, and symptoms; and physiological measures such as systolic blood pressure (SBP), diastolic blood pressure (DBP), heart rate (HR), breathing rate (BR), sympathetic activation (SA), parasympathetic activation (PSA), range of motion (ROM), and isokinetic peak torque (PT) in quadriceps and hamstrings at Week 0, 4, and 8. Finally, twenty-four patients were interviewed and informed about the rehabilitation program they would receive. As shown in Figure 1, the participants were allocated as follows: LBVG (n = 11) and nLBV (n = 13).”

For another example,

"2.7. Data Analysis

All data are reported as mean (SD). The sample size was determined using G*Power v 3.1.3, considering an a priori effect size f²(V) = .35 (medium size effect), α error probability = .05 and power (1 − β error probability) = .95. Although a sample size of 30 was recommended, the current sample included 24 participants. Based on the results of the Kolmogorov-Smirnov test, the non-parametric Mann-Whitney U test and Friedman test were used to examine the differences of variables between groups and to examine the changes of variables among times. Significance was set at P < .05. SPSS version 18.0 (SPSS Inc., Chicago, IL) was used for all analyses.

Results

3.1. Effect of LBV on psychological condition

The rehabilitation program applied in this study improved the psychological measures of LBVG and nLBVG. As shown in Table 3, although the anxiety was not significantly different between groups at all times, the pain of LBVG was significantly lower than that of nLBVG at Week 8 and the symptoms of LBVG were significantly lower than those of nLBVG at Week 4 and at Week 8. In other words, a significant effect of the LBV-intervention was found out concerning pain and symptoms.

Table 3. Differences and changes in psychological scales.

Item

(points)

Groups

Week

LBVG (n = 11)

nLBVG (n = 13)

Z (p)*

Pain

0

7.56 ± 1.21

7.70 ± 1.58

-0.511 (0.649)

4

4.10 ± 1.44

4.53 ± 1.20

-0.503 (0.649)

8

2.13 ± 0.90

4.54 ± 1.33

-3.661 (0.001)

X2 (p)**

20.150 (0.001)

17.280 (0.001)

Anxiety

0

6.39 ± 1.09

6.10 ± 1.24

-1.000 (0.361)

4

3.72 ± 0.64

4.08 ± 0.95

-1.193 (0.277)

8

3.01 ± 0.91

3.58 ± 0.95

-1.230 (0.252)

X2 (p)**

17.714 (0.001)

17.522 (0.001)

Symptom

0

7.04 ± 1.46

7.12 ± 1.89

-0.353 (0.733)

4

3.50 ± 1.16

6.71 ± 1.10

-4.119 (0.001)

8

2.31 ± 0.90

3.95 ± 1.21

-3.090 (0.002)

X2 (p)**

18.558 (0.001)

12.875 (0.002)

All data represents mean ± standard deviation. LBVG and nLBVG mean local body vibration group and non-local body vibration group, respectively. Symbols * and ** were analyzed by Mann-Whitney U test and Friedman test, respectively.

3.2. Effect of LBV on physiological condition

3.2.1. Effect of LBV on cardiorespiratory variables

The rehabilitation program applied in this study somewhat improved the cardiorespiratory measures of LBVG and nLBVG. As shown in Table 4, SBP, HR, and BR were significantly changed in the LBVG, whereas only SBP and BR were significantly changed in the nLBVG from the baseline to the end of experiment. However, all the cardiorespiratory measures were not significantly different between groups at all times. In other words, a significant effect of the LBV-intervention was not found out concerning cardiorespiratory variables of physiological condition.

Table 4. Differences and changes in cardiorespiratory variables.

Item

(units)

Groups

Week

LBVG (n = 11)

nLBVG (n = 13)

Z (p)*

Systolic blood

0

135.09 ± 5.17

134.69 ± 12.59

-0.351 (0.733)

pressure (mmHg)

4

127.09 ± 12.99

126.23 ± 7.17

-0.350 (0.733)

8

123.45 ± 5.63

124.31 ± 4.85

-0.235 (0.820)

X2 (p)**

13.905 (0.001)

7.714 (0.021)

Diastolic blood

0

80.55 ± 7.15

81.38 ± 12.49

-0.991 (0.331)

pressure (mmHg)

4

79.73 ± 7.73

79.08 ± 11.11

-0.872 (0.392)

8

79.09 ± 7.11

81.15 ± 11.01

-1.546 (0.134)

X2 (p)**

0.800 (0.670)

0.792 (0.673)

Heart rate

0

77.09 ± 6.47

71.08 ± 8.30

-1.887 (0.063)

(beats/min)

4

73.45 ± 6.22

78.62 ± 9.99

-1.457 (0.150)

8

69.09 ± 7.71

73.69 ± 7.22

-1.484 (0.150)

X2 (p)**

11.538 (0.003)

3.720 (0.156)

Breathing rate

0

29.61 ± 7.16

27.65 ± 5.54

-1.251 (0.228)

(reps.)

4

19.85 ± 4.19

19.14 ± 3.06

-0.611 (0.569)

8

19.60 ± 4.34

18.94 ± 3.48

-0.669 (0.531)

X2 (p)**

13.818 (0.001)

16.769 (0.001)

All data represents mean ± standard deviation. LBVG and nLBVG mean local body vibration group and non-local body vibration group, respectively. Symbols * and ** were analyzed by Mann-Whitney U test and Friedman test, respectively.

3.2.2. Effect of LBV on autonomic nervous system

The rehabilitation program in this study somewhat improved the measures of autonomic nervous system in LBVG and nLBVG. As shown in Table 5, SA and PA were significantly changed in the LBVG, whereas only SA was significantly changed in the nLBVG from the baseline to the end of experiment. Specifically, SA measure of LBVG was significantly different compared with that of nLBVG at Week 4. The PSA measure of LBVG was significantly higher than that of nLBVG at Week 4 and Week 8. In other words, a significant effect of the LBV-intervention was found out concerning decreased SA and increased PSA of physiological condition.

Table 5. Differences and changes in autonomic nerve system variables.

Item

Groups

Week

LBVG (n=11)

nLBVG (n=13)

Z (p)*

Sympathetic

0

7.61 ± 0.72

7.23 ± 0.51

-1.352 (0.186)

activation

4

6.32 ± 0.63

7.69 ± 1.15

-3.023 (0.002)

8

5.77 ± 0.57

6.46 ± 1.33

-1.257 (0.228)

X2 (p)**

14.727 (0.001)

7.882 (0.019)

Parasympathetic

0

3.55 ± 0.40

3.42 ± 0.38

-0.849 (0.424)

activation

4

4.49 ± 0.36

3.58 ± 0.40

-4.003 (0.001)

8

4.96 ± 0.28

3.50 ± 0.46

-4.162 (0.001)

X2 (p)**

22.000 (0.001)

2.923 (0.232)

All data represents mean ± standard deviation. LBVG and nLBVG mean local body vibration group and non-local body vibration group, respectively. Symbols * and ** were analyzed by Mann-Whitney U test and Friedman test, respectively.

3.2.3. Effect of LBV on strength and ROM

The rehabilitation program in this study improved the measures of isokinetic strength and ROM only in the LBVG. As shown in Table 6, the PTs of extensor and flexor were significantly changed in the LBVG, whereas those were not significantly changed in the nLBVG from the baseline to the end of experiment. Specifically, extensor PT of LBVG was significantly higher than that of nLBVG at Week 8. The flexor PTs of LBVG were also significantly higher than those of nLBVG at Week 4 and Week 8. The ROM of LBVG was also significantly higher than those of nLBVG at Week 4 and Week 8. In other words, a significant effect of the LBV-intervention was found out concerning increased isokinetic strength and improved ROM of an operated knee joint.

Table 6. Differences and changes in strength and ROM of knee joint.

Item

(units)

Groups

Week

LBVG (n=11)

nLBVG (n=13)

Z (p)*

Extensor

0

77.82 ± 12.91

70.00 ± 6.53

-2.311 (0.022)

peak torque

4

123.09 ± 35.18

100.15 ± 33.83

-1.574 (0.119)

(Nm)

8

178.73 ± 27.45

110.23 ± 53.88

-3.229 (0.001)

X2 (p)**

13.273 (0.001)

1.385 (0.500)

Flexor

0

52.36 ± 5.24

47.08 ± 10.47

-1.196 (0.252)

peak torque

4

63.91 ± 6.24

43.08 ± 16.27

-3.286 (0.001)

(Nm)

8

68.00 ± 10.26

50.23 ± 19.43

-2.052 (0.041)

X2 (p)**

17.636 (0.001)

3.231 (0.199)

Range

0

94.55 ± 16.05

86.00 ± 15.35

-1.279 (0.207)

of motion

4

121.73 ± 14.36

89.62 ± 17.47

-3.604 (0.001)

(°)

8

127.82 ± 12.46

92.38 ± 27.65

-2.970 (0.002)

X2 (p)**

19.860 (0.001)

1.385 (0.500)

All data represents mean ± standard deviation. LBVG and nLBVG mean local body vibration group and non-local body vibration group, respectively. Symbols * and ** were analyzed by Mann-Whitney U test and Friedman test, respectively.

: ”

#2. Comments and Suggestions: In my opinion there should have been 2 other arms to the study to convince me that VAST makes a difference - an arm with vibration and no music and an arm with "conservative " rehab ie no music or vibration.

#2. Response: The group treated by LBV was the experimental group, which received vibroacoustic pulses, whereas the control group received massages by a VAST device without vibroacoustic pulses. In other words, the study was treated with VAST in the experimental group, but not in the control group to investigate the effectiveness of VAST. In the manuscript, the corrected sentences as follows: " The LBVG received vibroacoustic pulses, whereas the nLBVG received massages by a VAST device without vibroacoustic pulses as a placebo.”

Finally, we got the English Editing Service from editors at Editage, a division of Cactus Communications. I attached "Certificate of English Editing” file.

Thank you so much.

Best regards,

Yong-Seok, Jee

Reviewer 2 Report

The material and method is scarcely explained, it is very advisable to follow the CONSORT guide for clinical trials. The measurements are very poorly explained and without references, it is not well understood what scale is used for anxiety and depression. Nor how to measure the ROM, without references to know if it is made with something reliable.   It is necessary to justify why the experimental group has more treatment, instead of using the device off. On the other hand, it is necessary to discuss the effect of having more treatment in the experimental group than in the control. It should also appear in the limitations section . There is no limitations section, which is mandatory.    

Author Response

Answers to reviewer’s comments 

Thank you for your kind advice and comments for publication in Medicina. We revised the manuscript as per your comments. We represented the specific modifications in response to the comments by blue-letters in our manuscript. We sincerely appreciate your comments because your comments make our manuscript better.

Reviewer 2:

#1. Comments and Suggestions: The material and method is scarcely explained, it is very advisable to follow the CONSORT guide for clinical trials.

#1. Response: As your recommend, we tried to re-explain on the material and method following the CONSORT guide.

For example,

"2. Materials and Methods

2.1. Study Design and Participants

This study was observed in a research center from December 1, 2017 to February 6, 2018. The first assessment was conducted from December 1 to 2, 2017, the second assessment from January 2 to 3, 2018, and the last assessment from February 5 to 6, 2018. Participants were recruited through the recommendation of a surgeon who understood LBV. Prior to the study, participants received detailed explanations regarding study procedures and were then asked to complete questions. The included criteria required that participants underwent an ACL reconstruction no more than seven days before study commencement. Participants were evaluated by clinical and radiological criteria, level of knee pain during the past month, and pain or difficulty in standing from a sitting position or climbing stairs [9]. All participants were patients who did not exercise regularly for over six months. Additionally, participants were also included if they had not received treatment/ medication for weight loss or anything known to affect body composition, and if they did not have any internal metallic materials. Participants’ mean (SD) age was 29.25 (14.51) years. After excluding three participants (one had cardiac surgery three years prior and two refused to participate because their homes were far from the hospital or research center) of fifty-five eligible participants, the remaining fifty-two participants belonged to one of two groups by lot and were randomly allocated to each group as shown in Figure 1. Of the 26 patients in the experimental group which allocated in the LBV group (LBVG), one did not receive the assessment, two lost in the follow-up phase, eight underwent arthroscopic surgery for complex injuries to their ACL or posterior cruciate ligament, and four underwent ACL arthroplasty. Therefore, eleven participants of the LBVG which underwent ACL reconstruction were analyzed in our study. Further, of the 26 patients in the control group which allocated in the non-LBV group (nLBVG), two did not receive the assessment, two lost in the follow-up phase, and nine underwent arthroscopic surgery with simple ACL rupture. Therefore, thirteen participants of the nLBVG which underwent ACL reconstruction were analyzed in our study. Excluded criteria had any history of impairment of a major organ system and psychiatric diseases. Participants with tumors, vascular inflammation, or kidney stones were also excluded. The LBVG received vibroacoustic pulses, whereas the nLBVG received massages by a VAST device without vibroacoustic pulses as a placebo. This study investigated psychological measures such as pain, anxiety, and symptoms; and physiological measures such as systolic blood pressure (SBP), diastolic blood pressure (DBP), heart rate (HR), breathing rate (BR), sympathetic activation (SA), parasympathetic activation (PSA), range of motion (ROM), and isokinetic peak torque (PT) in quadriceps and hamstrings at Week 0, 4, and 8. Finally, twenty-four patients were interviewed and informed about the rehabilitation program they would receive. As shown in Figure 1, the participants were allocated as follows: LBVG (n = 11) and nLBV (n = 13). All participants were assigned using random number tables and assigned identification numbers upon recruitment. Participants’ characteristics, which indicated homogeneity, are presented in Table 1.

Figure 1. Participant allocation (consolidated standards for reporting of trials flow diagram).”

#2. Comments and Suggestions: The measurements are very poorly explained and without references, it is not well understood what scale is used for anxiety and depression.

#2. Response: As your recommendation, we found the references and put them to the end of sentences by reference number.

For example,

“And the scales for measuring the levels of anxiety and symptoms were similar to the pain scale which used to Visual Analogue Scale. The scales for measuring the levels of anxiety and symptoms were similar to the pain scale. In other words, the anxiety or symptom scale ranged from no anxiety or no symptom (close to “0”) to severe anxiety or symptom (close to “10”) [10].”

[10] Jee, Y.S. The efficacy and safety of whole-body electromyostimulation in applying to human body: based 372 from graded exercise test. J Exerc Rehabil 2018, 14, 49-57.

#3. Comments and Suggestions: Nor how to measure the ROM, without references to know if it is made with something reliable.

#3. Response: As your recommendation, we found the references. We corrected the sentences and put them to the end of sentences by reference numbers.

" The ROM of an operated knee joint with ACL reconstruction was measured using a goniometer in the extended and flexed positions [31,32]. During the measurement, the ROM of the injured side was measured twice when participants were actively extended or flexed. Then, the mean values of the two measurements were recorded.”

[31] Ekhtiari, S.; Horner, N.S.; de Sa, D.; Simunovic, N.; Hirschmann, M.T.; Ogilvie, R.; Berardelli, R.L.; Whelan, D.B.; Ayeni, O.R. Arthrofibrosis after ACL reconstruction is best treated in a step-wise approach with early recognition and intervention: a systematic review. Knee Surg Sports Traumatol Arthrosc 2017, 25, 3929-3937.

[32] Kilinc, B.E.; Kara, A.; Celik, H.; Oc, Y.; Camur, S. Is anterior cruciate ligament surgery technique important in rehabilitation and activity scores? J Exerc Rehabil 2016, 12, 232-237.

In addition, we inserted the reference to the end of Isokinetic strength measurement as follows:

[33] Zhang, X.; Hu, M.; Lou, Z.; Liao, B. Effects of strength and neuromuscular training on functional performance in athletes after partial medial meniscectomy. J Exerc Rehabil 2017, 28, 13, 110-116.

#4. Comments and Suggestions: It is necessary to justify why the experimental group has more treatment, instead of using the device off. On the other hand, it is necessary to discuss the effect of having more treatment in the experimental group than in the control. It should also appear in the limitations section. There is no limitations section, which is mandatory.

#4. Response:

Thank you for the reviewer's findings. The answer is:

"The group treated by LBV was the experimental group, which received vibroacoustic pulses, whereas the control group received massages by a VAST device without vibroacoustic pulses. In other words, the study was treated with VAST in the experimental group, but not in the control group to investigate the effectiveness of VAST. Because we want to find out the effectiveness of VAST, the LBVG (experimental group) received vibroacoustic pulses, whereas the nLBVG (control group) received massages by a VAST device without vibroacoustic pulses as a placebo.

Finally, we got the English Editing Service from editors at Editage, a division of Cactus Communications. I attached "Certificate of English Editing” file.

Thank you so much.

Best regards,

Yong-Seok, Jee

Round 2

Reviewer 1 Report

The authors have addressed my major concerns by changing the study to deal with ACL reconstruction alone and I think this improves the value.

I believe there a few limitations that need to be mentioned 1 Fewer numbers than power study suggested 2 What is the clinical implication of the changes in para symph results at 4 nd 8 weeks, in other words does the stat sign results mirror the clinical changes 

Author Response

1st Reviewer’ Revision: I believe there a few limitations that need to be mentioned.

1) Fewer numbers than power study suggested.

2) What is the clinical implication of the changes in para symph results at 4 and 8 weeks, in other words does the stat sign results mirror the clinical changes.

Response:

1) Fewer numbers than power study suggested.

As reviewers point out, stat power was not met by selecting only ACL reconstruction patients among patients who had knee surgery. To counteract this, statistical methods were analyzed through nonparametric comparisons. Thank you for improving the quality of this study due to the reviewer's suggestion and comment.

2) What is the clinical implication of the changes in para symph results at 4 and 8 weeks, in other words does the stat sign results mirror the clinical changes.

The reason for observing the change in variables by 0, 4, and 8 weeks was that several studies reported that psychophysiological changes were found in four weeks of rehabilitation, and that some studies showed psychophysiological changes in eight weeks of rehabilitation. We analyzed and evaluated the changes in the rehabilitation period. Thank you for improving the quality of this study due to the reviewer's suggestion and comment.

Reviewer 2 Report

The authors fulfilled the requested changes. Congratulations for the manuscript

Author Response

2nd Reviewer’ Revision

I don't feel qualified to judge about the English language and style.

Response:

This paper was revised by a native English editor, again. 

Thank you for improving the quality of this study due to the reviewer's suggestion and comment.
